# Women Facing Psychological Abuse: How Do They Respond to Maternal Identity Humiliation and Body Shaming?

**DOI:** 10.3390/ijerph18126627

**Published:** 2021-06-20

**Authors:** Marina B. Martínez-González, Diana Carolina Pérez-Pedraza, Judys Alfaro-Álvarez, Claudia Reyes-Cervantes, María González-Malabet, Vicente Javier Clemente-Suárez

**Affiliations:** 1Department of Social Science, Universidad de la Costa, Barranquilla 080001, Colombia; dperez26@cuc.edu.co (D.C.P.-P.); alfajudys@gmail.com (J.A.-Á.); creyes3@cuc.edu.co (C.R.-C.); 2Department of Political Sciences and International Affairs, Universidad del Norte, Barranquilla 080003, Colombia; malabetm@uninorte.edu.co; 3Faculty of Sport Sciences, Universidad Europea de Madrid, 28670 Villaviciosa de Odón, Spain; vctxente@yahoo.es; 4Grupo de Investigación en Cultura, Educación y Sociedad, Universidad de la Costa, Barranquilla 080001, Colombia

**Keywords:** domestic violence, coping strategies, early maladaptive schemes, motherhood, body shaming

## Abstract

This research analyzes the decisions made by women facing simulated situations of psychological abuse. Seventy-three women (36.9 ± 13.6 years) who had been victims of domestic violence participated. The analysis was based on their coping strategies, early maladaptive schemes, and their decisions in response to vignettes describing the following domestic violence situations: humiliation to women’s maternal identity with children as witnesses and body shaming. We used Student’s t and Mann–Whitney tests to compare the results between groups. The participants presented some coping strategies (social support seeking, wishful thinking, and professional support seeking) and several early maladaptive schemes (emotional deprivation, defectiveness/shame, social isolation/alienation, failure to achieve, attachment, and subjugation) associated with their reactions facing a situation of humiliation with children as witnesses. When the humiliation was against the body image, their reactions were associated with some coping strategies (wishful thinking, professional support seeking, autonomy, negative auto-focus coping, and positive reappraisal) and one maladaptive scheme (defectiveness/shame). Women who reacted avoidantly showed higher social and professional support seeking but experienced higher indicators of discomfort and deterioration of self-esteem than those who opted for assertive decisions. The presence of children as witnesses seems to be a factor of stress in the configuration of coping strategies and maladaptive schemes in female victims of domestic violence. The evolution of early maladaptive schemes and coping strategies requires observation to avoid the risk of isolation and permanence in victimizing relationships.

## 1. Introduction

Domestic violence has transcended from the private to the public sphere and constitutes a human rights problem where women have a greater probability of being victims than men [1,2]. Situations of psychological, physical, and sexual abuse against women are often studied, mainly associated with intimate partner violence [3,4]. Nevertheless, in every kind of abuse, psychological violence is imminently present through several manifestations of isolation, distortion of reality, intimidation, emotional abuse, harassment, and humiliation [5,6]. This kind of violence against women is not always perceived because it is masked as cultural patterns of family socialization and traditional interactions, according to each context [7,8]. In many cases, shame and humiliation are used as a control and manipulation strategy, affecting family interactions and expectations about relationships over time in society [9].

There are several reasons why a woman remains in a violent relationship [3,10,11], but one of the most important is the presence of children born from the couple’s union [12]. Some studies showed that most cases of gender violence are present in families with kids [2,13]. Being humiliated about maternal abilities or body image in public and private spheres is one of the main ways of intimate partner victimization [14]. The process of motherhood implies developing a new identity for women. This new identity demands an emotional process, in which can appear anxiety, fear, shame, and thoughts of inability to take care of a child until the woman recognizes herself as an empowered mother [15,16]. Perpetrators take advantage of the mother’s identity as a caregiver, her relationship with the children, and her desire to protect them as a way to victimize her [17]. The main reason women allow violence is to ensure the children’s wellness, keeping the family together, and growing in a biparental family [5]. Nevertheless, children that grow in this environment are exposed to the same violence and stress as the mother [18]. Children’s emotional, behavioral, social, academic achievement and future interpersonal performance are negatively affected by the conflict between parents [19,20,21], highlighting the necessity of prevention, not only as a form of protection for women but also for the kids [22].

Women involved in domestic violence experience anxiety episodes, stress, and depression because they are continuously exposed to pressure, control, manipulation, and coercion, from their intimate partners that want to dominate and subdue [4,6,7,23,24,25]. Female victims of body shaming usually present low self-esteem, self-efficacy, and damaged self-concept [26,27]. Regarding psychological violence, body shaming involves offenses that negatively influence the body image and women’s belief in their capacity to achieve their desired outcomes successfully [26,27].

The psychological and social consequences for these women later in life include the use of psychoactive substances, isolation from social life, and low self-esteem [26]. When victims start suffering abuse, they leave their social support system, increasing their vulnerability [28,29]. This situation produces in women hopelessness, helplessness, fear, pain and grief [28,29,30]. Those thoughts and feelings impact the escalation of conflicts and their implications in worsening situations [30,31]. Previous research postulated that some female victims of domestic violence, who are terrorized in their houses, can fight back physically as an instinctive reaction and as a defense mechanism. These women are resistant to violence to scope fatal consequences or unbearable situations [32]. The literature shows that some women who sought assistance in the face of violence also reported violent behaviors against their controlling partner [33,34,35]. This tendency for violence shows the need to analyze the coping strategies and cognitive schemes, resulting in the escalation of partner conflict towards domestic violence. Nevertheless, the cognitive profile that defines how women face domestic violence might force them to remain in an abusive situation and increase the risk of victimization.

The cognitive profile is a construct that considers the evolution of early maladaptive schemes and coping strategies, which are a product of the relationship between the behavior and the environment [36]. Early maladaptive schemes are self-sabotaging and dysfunctional patterns of thought that start in childhood, generated in the relationship with one or both parents, that remain throughout life [37]. These schemes originate from real situations of daily life and can trigger adaptive responses. However, they become maladaptive in adulthood when the cognitive representations and behaviors do not fit reality. In intimate violent relationships, the schemes might be related to a form of attachment that transforms into a low capacity for conflict resolution. Thus, maintaining violent situations through increasing threats [38,39] is also associated with maladaptive cognitive patterns in interpersonal relationships [40].

Meanwhile, coping strategies are cognitive-behavioral processes that aim to solve the demands of the internal and external environments, when there is an overflow psychological resources, and might be oriented towards problem-solving or focused on emotional regulation [41]. Thus, coping implies conscious efforts to manage the problems created by stressful events exceeding the individual’s capacity and resources [40]. Coping strategies are considered adaptive when the strategies include observable behaviors such as receiving support from others and acting to overcome the stressful situation. However, when the strategies are unobservable and imply emotional efforts such as denial or self-distraction to avoid stress, coping is considered maladaptive [42].

Research findings among female victims of abuse reported a positive association between adaptive coping and psychological well-being, while studies associated maladaptive coping with psychological difficulties and lack of perceived social support [43]. Other studies found a relationship between strategies such as escape-avoidance or positive re-evaluation, with high rates of abuse exposure, as well as with high prediction of the development of some mental disorder [25,27,44,45,46,47]. Thus, female victims of violence generally use passive coping strategies such as wishful thinking, seeking social support, and emotional avoidance to face the abusive situation [48]. However, women who use confrontation and aggression to solve conflicts, instead of leaving the relationship, are at higher risk of suffering domestic violence [49]. Early maladaptive schemes and coping strategies are two main factors that affect personal decisions. Poor coping strategies or non-adaptative early schemes could produce aggressive, submissive, or avoidant behaviors that would reinforce the family violence context, facilitating abuse and gender violence [37,38,41,42].

However, studies about women’s sense of resilience and coping showed attitudes that played an essential role in helping women endure these difficulties—attitudes such as professional development, making sense of the experience, help-seeking, and focusing on children [50]. Nevertheless, there is little evidence of humiliation in front of children as a risk factor for coping and maladaptive schemes related to domestic violence. It is important to understand that children witnessing violent situations becomes a stress factor for women and triggers negative coping strategies for facing these situations. Today, studies about the relationship between stress and coping strategies analyze the role of transactional processes in adaptation and how these variables mutually influence each other [50]. The study results about partner violence contribute to developing comprehensive interventions that focus on reducing clinical rates, pathological stress, maladaptive schemes, and address context-related factors, which make the effects last in the long term.

Therefore, this study aims to analyze the coping strategies and early maladaptive schemes of domestic violence victims facing psychological abuse about women maternal identity with children as witnesses and body shaming. Both violent situations are stressful for women, manifesting in a maladaptive cognitive profile. We hypothesized that (i) demographic variables would affect coping strategies and early maladaptive schemes of female victims of domestic violence, (ii) coping strategies and early maladaptive schemes will differ depending on the decision made by women facing psychological abuse with children as witnesses, and (iii) coping strategies and early maladaptive schemes will differ depending on the decision made by women facing psychological abuse of body shaming.

The Caribbean region of Colombia presents high levels of domestic violence. In 2019, 6330 cases were registered among the following seven territories: Atlántico (2016), Bolívar (1199), La Guajira (343), Cesar (695), Córdoba (565), Magdalena (952) and Sucre (560) [51]. Analyzing how female victims of domestic violence deal with psychological abuse from their partners and how the decision of these women is related to coping strategies and early maladaptive schemes can help strengthen intervention programs to prevent and mitigate the effects of violence. Furthermore, this research is unique because we examine, on one side, the humiliation of maternal identity with children as witnesses and, on the other side, body shaming. Neither of these phenomena have been studied with the specific situation and context that we are proposing.

## 2. Materials and Methods

### 2.1. Participants

A total of 73 female victims of domestic violence participated. We approached participants from the seven major cities in the Caribbean coastal region of Colombia. We used purposive sampling through an organization [52] using proportional quotas [53] for each city. We assessed the women through the cooperation of seven foundations that support women to overcome domestic violence. Initially, we searched foundations on the Internet; we contacted them and had a meeting with their leaders to present the project and its benefits. Those who were interested in the proposal signed a cooperation agreement. They shared the information with their beneficiaries and, finally, those who agreed to be volunteers participated.

The majority of the participants were married or in a relationship (65.8%). Most of them only finished high school (41.7%), some women completed technical studies (23.6%), some of them obtained a professional degree (18.1%), and a few only finished primary school (16.7%). Their mean age was 36.9 (SD = 13.62). Their employment status was unemployed (47.9%), employed (31.5%), and freelance (20.5%).

### 2.2. Procedure

The present study presents laboratory research, relying on the use of the vignette methodology that is mainly used for assessing attitudes, norms, and perceptions about sensitive topics such as violence [54,55,56].

The research was conducted from August 2017 to July 2019. We cooperated with the foundations to complete the sample. We informed all the participants about the study’s aims and procedures, and they filled in the informed consent form. After this, we implemented the paper and pencil method using copies of questionnaires distributed by a psychologist from the research team, who was previously trained on the data gathering process.

The participants answered the following questionnaires:

Vignettes to present two hypothetical situations that represent cases of psychological abuse. The first situation describes a humiliation situation incorporating children as witnesses: “Your partner is taking a nap. The children enter his room, start to play; they begin making noise and wake him up. Your partner calls you out loud, very upset, and begins to tell you that you are useless, that you are a disaster as a mother and wife, and that your role has grown too big for you”. The second situation describes a situation of body shaming: “You are preparing to attend a party with your partner. Days ago, you bought a dress that you liked, and you decided to wear it for this event. Seeing you in your new dress, your partner makes a surprising and displeasure gesture. He says that dress looks terrible on your body and that you should take it off”. After every described situation, the participant read the question, “What would you do?”.

We registered the decisions taken in each vignette using the following categorical scale that includes the response options: assertive, avoidant, aggressive, submissive, and supportive. For situation 1: you ignore their comments and take the children out (avoidant); you talk with someone and ask for help to calm your husband (supportive); you tell him that children play, and if they are noisy, it does not mean you are a bad mother (assertive); you tell him he is a bad father who does not care about his children (aggressive); you punish the children for waking up their father (submissive). For situation 2: you ignore his comments and keep the dress (avoidant); you tell someone else to convince him to change his mind (supportive); you explain that you feel good in that dress and you do not want to change it (assertive); you tell him that he does not look good either and ridicule him for what he told you (aggressive); you take the dress off without arguing (submissive).

After completing the vignettes, the participants responded to the following scales.

First, the modified coping strategies scale [57] evaluates coping strategies and consists of 98 items distributed in these factors: problem-solving coping, social support seeking, wishful thinking, religious coping, emotional distancing, professional support seeking, overt emotional expression, avoidance coping, positive reappraisal, expression of coping difficulty, negative auto-focused coping, and autonomy coping. The instruction for answering this scale are: Below, you will find phrases that a person could use to describe himself. Please read each sentence and decide how accurately it describes you. When you are not sure, base your answer on what you feel emotional, not on what you think is true. Choose the highest value from 1 to 6 that describes you and mark it next to each phrase. Response options are Likert 1–6 (1 = never; 2 = almost never; 3 = sometimes; 4 = frequently; 5 = almost always; and 6 = always). Cronbach’s alpha for this scale in this study was 0.887. The evaluation for research purposes calculates the total score by subscale [57]. The score interpretation for this scale is [57]:Problem-solving: low (28; percentile 25); medium (33; percentile 50) and high (39; percentile 75);Social support seeking: low (18; percentile 25); medium (22; percentile 50) and high (29; percentile 75);Wishful thinking: low (17; percentile 25); medium (22; percentile 50) and high (27; percentile 75);Religious coping: low (14; percentile 25); medium (21; percentile 50) and high (27; percentile 75);Emotional distancing: low (19; percentile 25); medium (24; percentile 50) and high (30; percentile 75);Professional support seeking: low (6; percentile 25); medium (10; percentile 50) and high (15; percentile 75);Overt emotional expression: low (9; percentile 25); medium (12; percentile 50) and high (16; percentile 75);Avoidance coping: low (13; percentile 25); medium (15; percentile 50) and high (19; percentile 75);Positive reappraisal: low (15; percentile 25); medium (18; percentile 50) and high (22; percentile 75);Expression of coping difficulty: low (11; percentile 25); medium (15; percentile 50) and high (19; percentile 75);Negative auto-focused coping: low (6; percentile 25); medium (8; percentile 50) and high (11; percentile 75);Autonomy: low (5; percentile 25); medium (7; percentile 50) and high (9; percentile 75).

Second, the Young schema questionnaire short form (YSQ-SF) [58] is a self-applied questionnaire oriented toward evaluating early maladaptive schemes. It is formed by 75 items distributed in 15 factors: emotional deprivation, abandonment/deprivation, mistrust/abuse, defectiveness/shame, social isolation/alienation, failure to achieve, dependency/incompetence, vulnerability to harm, attachment, subjugation, self-sacrifice, emotional inhibition, unrelenting standards, entitlement/grandiosity, and insufficient self-control. The instructions for answering this scale are: Below are different ways people can cope with stressful problems or situations that arise in life. The ways of coping described here are neither good nor bad, nor better or worse. Some people simply use some forms more than others, depending on the problem situation. Try to remember the different stressful situations or problems experienced in recent years and respond by putting an X in the column that points to the arrow, the number that best indicates how common this form of behavior has been in stressful situations. Response options are Likert 1–6 (1 = totally false; 2 = most of the false times; 3 = more false than true; 4 = more true than false; 5 = most times true and 6 = describe me perfectly). Cronbach’s alpha for this scale in this study was 0.95. The evaluation for research purposes calculates the total score by subscale. The minimum possible score is 5 (absence of the scheme) and the maximum possible score is 30 (presence of the scheme) [58].

At the end of the study, the research team offered a workshop and presented a research report to generate an appropriation process of the research results for each participating foundation. Likewise, we provided recommendations for the development of positive coping strategies and resilient attitudes in female victims of domestic violence.

### 2.3. Data Analysis Strategies

We used the JASP^®^ Statistics software (The JASP Team, Amsterdam, The Netherlands) to analyze the data. First, we analyzed the distribution of the participants considering their decisions to face the proposed situations. When they answered the abuse situation with children as witnesses, we found that the majority opted for the assertive option (61.64%), followed by those who assumed an avoidant behavior (28.77%). Only a small group opted for an aggressive reaction (8.22%). Just one woman decided to seek help and none chose a submissive position. In the second situation, body shaming, the highest proportion of the participants choose the assertive (65.75%) and the avoidant attitudes (24.66%). Just a small proportion tended to select the submissive position (6.85%) but the aggressive reaction and seeking help were only chosen by one woman, respectively (see Table 1). Considering this, we explored the differences in coping strategies and early maladaptive schemes by depending on the participants’ reactions in two groups: the women who opted the assertive option and the women who decided on the avoidant option.

To compare the main differences for coping strategies and early maladaptive schemes between groups (assertive or avoidant), we used the Student’s *t*-test. We also used the Mann–Whitney test due to some results showed normality deviation. The level of significance for all the comparisons was set at *p* ≤ 0.05. For the Student’s *t*-test, the effect size is given by Cohen’s d, and for the Mann–Whitney test, by the rank biserial correlation. MANOVA was also used to analyze the possible effects of demographic variables such as age group, marital status, educational level, and employment status on early maladaptive schemes and coping strategies.

## 3. Results

No significant differences were found for the coping strategies and early maladaptive schemes scores related to age—Wilks’ Λ = 0.17, F (81, 129) = 1.29, *p* = 0.096, η2 = 0.45; marital status—Wilks’ Λ = 0.09, F (108, 169) = 1.31, *p* = 0.06, η2 = 0.45; educational level—Wilks’ Λ = 0.03, F (162, 243) = 1.23, *p* = 0.07, η2 = 0.44; nor employment status—Wilks’ Λ = 0.33, F (54, 88) = 1.21, *p* = 0.21, η2 = 0.43.

### 3.1. Situation 1. Humiliation with Children as Witnesses

The test of normality (Shapiro–Wilk) showed significant results (*p* < 0.001) for four coping strategies, wishful thinking, autonomy, social support seeking, and professional support seeking, suggesting a deviation from normality. In the early maladaptive schemes, the normality test showed significant results (*p* < 0.001) for emotional deprivation, defectiveness/shame, social isolation/alienation, failure to achieve, attachment, and subjugation. The equality of variances (Levene’s) was no significant for coping strategies but showed significant results for the early maladaptive schemes of failure to achieve (*p* = 0.027) and attachment (*p* = 0.025).

The coping strategies with high scores were religious coping and problem solving. The medium scores were present in coping strategies such as avoidance, positive reappraisal, and emotional distancing. Low scores were related to expression of coping difficulty, autonomy, negative auto-focused coping, and overt emotional expression. Differences across decision groups were significant for three coping strategies: wishful thinking (*p* = 0.003) obtained a high/medium score for avoidant women and low/medium for assertive ones; social support seeking (*p* = 0.012) was medium for avoidant women and low/medium for assertive women; finally, professional support seeking (*p* = 0.032) was high/medium score for avoidant women and medium for the assertive ones.

In the case of early maladaptive schemes, the results showed that the participants did not have indicators with clinical significance. However, the schemes were present in these female victims, especially self-sacrifice, which had the highest score among all participants. The women that responded avoidantly tended to present slightly higher results than the assertive ones.

The hypothesis tests that we used showed significant differences across the groups in several mechanisms: abandonment/deprivation (*p* = 0.027), mistrust/abuse (*p* = 0.040), defectiveness/shame (*p* = 0.011), failure to achieve (*p* = 0.044), dependency/incompetence (*p* = *0*.008), attachment (*p* < 0.001), emotional inhibition (*p* = 0.018), and entitlement/grandiosity (*p* = 0.047). For those subscales, both the coping strategies and the early maladaptive schemes had higher scores for the women who decided on the avoidance option to face psychological abuse with children as witnesses (see Table 2). The above indicates that women who prefer to avoid confrontation in these types of situations would be more willing to seek social and professional help than those who handled the situation assertively, but they would also be more likely to wait for the situation to improve over time. However, they also experience higher indicators of discomfort and deterioration of self-esteem than those who opted for proactive decisions.

### 3.2. Situation 2. Body Shaming

The test of normality (Shapiro–Wilk) showed significant results for several coping strategies, wishful thinking (*p* = 0.002), professional support seeking (*p* < 0.001), overt emotional expression (*p* = 0.011), avoidance coping (*p* = 0.014), positive reappraisal (*p* = 0.043), negative auto-focused coping (*p* = 0.035) and autonomy (*p* = 0.005) suggesting a deviation from normality. The test also showed significant results for several early maladaptive schemes such as emotional deprivation (*p* = 0.005), abandonment/deprivation (*p* = 0.002), defectiveness/shame (*p* < 0.001), social isolation/alienation (*p* = 0.003), failure to achieve (*p* < 0.001), dependency/incompetence (*p* = 0.014), attachment (*p* = 0.001), subjugation (*p* < 0.001), emotional inhibition (*p* = 0.001) and insufficient self-control (*p* = 0.034). The equality of variances (Levene’s) was significant for both overt emotional expression (*p* = 0.028) and negative auto-focused (*p* = 0.007) coping strategies, and for two early maladaptive schemes, this was failure to achieve (*p* = 0.010) and attachment (*p* = 0.018).

The coping strategy with the highest score was religious coping. Medium scores were found for coping strategies such as problem solving, wishful thinking, emotional distancing, avoidance, and negative auto-focused coping. Low scores were found for expression of coping difficulty and autonomy.

Significant differences across decision groups were found for the following coping strategies: social support seeking (*p* = 0.041) yielded a medium score for avoidant and low/medium for assertive participants; professional support seeking (*p* = 0.015) was medium/high for avoidant and medium for assertive ones; overt emotional expression (*p* = 0.014) was medium/high for avoidant and low for assertive participants; and finally, expression of coping difficulty (*p* = 0.018) was low/medium for avoidant and medium for assertive women. In the case of early maladaptive schemes, the test just showed significant differences for defectiveness/shame (*p* = 0.038); nevertheless, the results are not clinically significant. For those subscales, both the coping strategies and the early maladaptive schemes had higher scores in the women who selected the avoidant behavior to face the situation of body shaming (see Table 3). As the previously evaluated situation showed, the women who tend to be avoidant would be more willing to seek social and professional help than those who assertively handed the situation. However, they would more likely react aggressively and experience difficulty in handing the situation, experiencing feelings of being overwhelmed. They also experience higher indicators of defectiveness when facing the body-shaming situation than women who react assertively.

## 4. Discussion

This study aimed to analyze women’s decisions when facing situations of psychological abuse and how these explain their coping strategies and early maladaptive schemes. Hypothesis (i) was not accomplished since demographic variables did not have a significant effect on coping strategies and early maladaptive schemes of female victims of domestic violence. Hypothesis (ii) was partially accomplished since some coping strategies (wishful thinking, social support seeking, and professional support seeking) and early maladaptive schemes (abandonment/deprivation, mistrust/abuse, defectiveness/shame, failure to achieve, dependency/incompetence, attachment, emotional inhibition, and entitlement/grandiosity) differed depending on the decision made by women facing psychological abuse with children as witnesses. Hypothesis (iii) was partially accomplished because just early maladaptive schemes (social support seeking, professional support seeking, overt emotional expression, and expression of coping difficulty), and one early maladaptive scheme (defectiveness/shame) were different depending on the decision made by women facing body shaming.

We found that age, marital status, educational level, and employment status were not significant factors to delimit coping strategies and maladaptive schemes in female victims of domestic violence who face different psychological abuse situations. Other studies found that demographic factors such as educational level and women’s employment status are relevant for the risk of victimization [10,11]. Furthermore, the marital status and having children with the offender increase the risk of remaining in a situation of domestic violence [2,12,13]. The risk appeared to be higher in older women who married at younger ages [11]. However, our results did not evidence differences between age groups for coping strategies and maladaptive schemes of female victims. A possible explanation for these findings is that many societies do not perceive psychological abuse as such because it is masked as a cultural pattern of family socialization and traditional interaction in many societies [7,8], and these psychological abuses are rooted from an early age in maladaptive beliefs and distorted relationship patterns. Another evidence for this is the participants’ tendency to react assertively or be avoidant to psychological abuse. It might signify a sign of relative tolerance for this type of aggression, which supports previous studies that state that avoidance and silence are the principal behaviors for facing domestic violence [59]. Additionally, an explanation for the absence of differences related to demographic characteristics would be that most participants were married or in a relationship, had a low educational level, and were unemployed. The size of the groups of women with different characteristics was small. These demographics are similar to the women population in the Colombian Caribbean region.

Concerning the presence of children as witnesses of the mother’s victimization, our results support other studies that indicate that the presence of children had a significant impact on the permanence of women in victimizing relationships [2,13,22]. Women evade abuse by accepting all the requests made by their partner, abandoning their wishes, and reinforcing their partner’s authority [60,61,62]. However, in this study, no woman reacted by directing punishment towards the children to surrender to the partner’s aggression. Additionally, a positive finding appears related to the wishful thinking coping strategy that showed low scores in women who reacted assertively, indicating that these women avoid delaying the conflict resolution. Some women were more convinced of their abilities to manage the conflict and reaffirm their resolution skills through the situation without resorting to violence. However, previous research has found that women often decided to handle violence alone because they do not trust social support or feel shame for social stigmatization. In this case, the avoidance coping strategy accounted for women’s depressive symptoms [47]. Hence, self-sufficiency needs to be carefully approached as it could represent a risk for the victims, giving them a false perception of control of the personal and situational threat, which leads the victim to remain in the conflict until it escalates to higher risk [62,63]. Staying in a relationship marked by frustration, self-sacrifice, and the couple’s lack of retribution might lead to aggressions from the woman to her partner in violent emotional explosions [30]. This aggression is most likely to happen when the situation threatens women’s maternal identity [15,16,49].

When facing humiliation based on body shaming, the overt emotional expression and the expression of coping difficulty were higher in women who reacted avoidantly. This kind of violence implies severe consequences for women’s self-confidence and self-esteem [26,27]. In addition, it is associated with feelings of impotence and deterioration of self-efficacy that could explain our findings [27]. As a positive result, these women also presented the highest levels of seeking social and professional support. Although limited, this is an important finding since these women are even more social-oriented than assertive, which means that they are open to receiving help from others to face this abusive situation. New related studies have found that the body plays a role in transforming women’s responses to violence [64,65,66]. They explored social interactions that shape the physical and emotional states through women’s bodies and embodiment in different contexts, including domestic violence. Social and cultural learnings generate somebody’s particular experience. These learnings, and their implications for female bodies, influence the behavior in an intimate relationship and trigger, intentionally or unconsciously, the response to a violent situation [64].

Recognizing the role of the body in responses to violence helps to understand women’s experiences. Considering the body and its relationship with self-concept has relevance for studies on female victims’ seeking help, the use of violence by women, and transformations in women’s responses to violence. Furthermore, exploring the body’s experience is an opportunity for female empowerment and the transformation of cultural conceptions about victims of violence [64,65,66].

Moreover, female victims that answered assertively facing this situation presented low tendencies to seek social and professional support. Women who consider themselves able to handle the situation alone might be at risk of being victimized for a long time [43] and tend to use more passive coping styles [21,43,48], such as denial, for emotional relief [62]. Usually, victims use denial to reduce the perception of violent acts and their consequences. In this sense, victims in denial or silence do not appreciate the magnitude of the risk they are in, much less the need to ask for help [60]. Silence and loneliness require attention. Isolation is one of the characteristics that leaves female victims without a support network and increases mental illness risk [43,47]. For this reason, professionals who attend to female victims in care services need to consider that these women have critical sensitivity, and professionals’ reactions are probably the first support for them, determining the decisions they will take immediately after [67]. Quality in care services, oriented toward restoring female victims and their children’s rights, is an essential tool to eradicate this social problem [67].

This study shows that active or conflict-focused strategies might be more favorable for psychological well-being than passive and emotional strategies, supporting the findings of previous research [43,48]. Hence, social support tends to exert a protective function and motivate women to leave abusive relationships [7,68]. Our findings have implications for improving social and professional support services and, consequently, for women’s psychological well-being. Understanding the involvement of women with children when facing domestic violence, as well as the body experience and self-concept, has the potential for sustained actions that allow the cognitive, behavioral, and affective change in the management of domestic violence [69].

### Limitations and Suggestions for Future Studies

This study explored the reaction of women who were victims of domestic violence to hypothetical situations of psychological abuse by their partners, presenting the presence of children in the situation as a stimulus. However, we did not control the number of children. Future studies should deepen the cognitive schemas and the coping of female victims with maternity and children’s specific conditions to broaden the understanding of the phenomenon.

The results of this study have limited generalization because it uses a non-probabilistic sample. Indeed, we did not balance the participants according to their rural or urban origins, which means that specific attitudes and views on gender cultural issues were overlooked. However, this study’s findings may help other research locations and generate new questions to explore this phenomenon and improve the support service for female victims of domestic violence.

This research highlighted that seeking social and professional support was significantly higher in those who reported earlier maladaptive schemes. We recruited participants from foundations that provide support to victims of domestic violence. Hence, it is pertinent to study if those women considered physical victimization as the principal motivation for help-seeking, or if they did not seek help when they experienced psychological abuse because they tolerate this type of violence. This is worrisome, as these women tend to overlook psychological abuse, which is the starting point for the escalation of violence.

Finally, the actual COVID-19 pandemic is a great contextual stressor [70,71,72,73,74,75,76,77] that could affect women abuse, then it would be interesting to analyze the impact of COVID-19 pandemic and lockdown in women abuse. 

## 5. Conclusions

Female victims of domestic violence presented some coping strategies (social support seeking, wishful thinking, and professional support seeking) and several early maladaptive schemes (emotional deprivation, defectiveness/shame, social isolation/alienation, failure to achieve, attachment, and subjugation) associated with their reactions facing a situation of humiliation with children as witnesses, which may indicate more difficult coping in this situation. In an abusive case of body shaming, they presented some coping strategies (wishful thinking, professional support seeking, autonomy, negative auto-focus coping, and positive reappraisal), but just one maladaptive scheme (defectiveness/shame) associated with their reaction. Demographic variables did not define their decisions when facing situations of psychological abuse. The configuration of different coping strategies and maladaptive schemes requires observation to avoid the risk of isolation and women’s permanence in victimizing relationships.

## Figures and Tables

**Table 1 ijerph-18-06627-t001:** Frequencies for reactions facing simulated situations of psychological abuse.

Situation	Reaction	Frequency	Percent	Accumulative Percent
1. Humiliation with children as witnesses	Aggressive	6	8.22	8.22
	Assertive	45	61.64	69.86
	Avoidant	21	28.77	98.63
	Seeking help	1	1.37	100.00
	Missing	0	0.00	
	Total	73	100.00	
2. Body shaming	Aggressive	1	1.37	1.37
	Assertive	48	65.75	67.12
	Avoidant	18	24.66	91.78
	Seeking help	1	1.37	93.15
	Submissive	5	6.85	100.00
	Missing	0	0.00	
	Total	73	100.00	

Note: The results of women who chose the aggressive response, help-seeking, and submission were not included in further analyses due to the low number of observations making them unfeasible.

**Table 2 ijerph-18-06627-t002:** Comparatives for coping strategies and early maladaptive schemes scores across the decisions facing humiliation with children as witnesses.

Scales	Subscales	Group	Number	Mean	Standard Deviation	Test	Statistic	Degrees of Freedom	*p*	Effect Size
Coping Strategies	Problem-solving	Assertive	45	34.600	7.551	Student	0.317	63	0.752	0.085
		Avoidant	20	33.900	9.602	Mann–Whitney	462.500		0.864	0.028
	Social support seeking *	Assertive	45	19.733	6.930	Student	−2.555	63	0.013	−0.687
		Avoidant	20	24.700	7.888	Mann–Whitney	274.000		**0.012**	−0.391
	Wishful thinking *	Assertive	45	19.178	9.061	Student	−2.572	63	0.012	−0.691
		Avoidant	20	25.050	7.015	Mann–Whitney	244.000		**0.003**	−0.458
	Religious coping	Assertive	45	26.489	8.050	Student	−0.695	63	0.490	−0.187
		Avoidant	20	28.000	8.189	Mann–Whitney	424.000		0.717	−0.058
	Emotional distancing	Assertive	45	22.978	7.408	Student	−0.386	63	0.701	−0.104
		Avoidant	20	23.750	7.552	Mann–Whitney	395.500		0.442	−0.121
	Professional support seeking *	Assertive	45	10.600	5.462	Student	−2.165	63	0.034	−0.582
		Avoidant	20	14.000	6.641	Mann–Whitney	299.500		**0.032**	−0.334
	Overt emotional expression	Assertive	45	10.800	4.165	Student	0.000	63	1.000	0.000
		Avoidant	20	10.800	4.491	Mann–Whitney	454.500		0.954	0.010
	Avoidance coping	Assertive	45	15.244	5.113	Student	−1.263	63	0.211	−0.340
		Avoidant	20	16.900	4.278	Mann–Whitney	320.000		0.065	−0.289
	Positive reappraisal	Assertive	45	18.822	4.554	Student	0.637	63	0.527	0.171
		Avoidant	20	18.050	4.419	Mann–Whitney	490.000		0.573	0.089
	Expression of coping difficulty	Assertive	45	10.533	3.514	Student	−0.302	63	0.764	−0.081
		Avoidant	20	10.850	4.694	Mann–Whitney	440.500		0.898	−0.021
	Negative auto-focused coping	Assertive	45	7.822	2.552	Student	1.147	63	0.256	0.308
		Avoidant	20	7.050	2.395	Mann–Whitney	520.500		0.315	0.157
	Autonomy	Assertive	45	5.111	2.707	Student	−0.840	63	0.404	−0.226
		Avoidant	20	5.700	2.364	Mann–Whitney	370.000		0.253	−0.178
Early Maladaptive Schemes	Emotional deprivation*	Assertive	45	11.311	5.888	Student	−0.636	63	0.527	−0.171
		Avoidant	20	12.300	5.536	Mann–Whitney	398.500		0.466	−0.114
	Abandonment/Deprivation	Assertive	45	11.800	5.667	Student	−2.260	63	**0.027**	−0.607
		Avoidant	20	15.300	5.975	Mann–Whitney	291.000		0.024	−0.353
	Mistrust/Abuse	Assertive	45	12.556	4.143	Student	−2.101	63	**0.040**	−0.565
		Avoidant	20	15.050	5.000	Mann–Whitney	329.500		0.087	−0.268
	Defectiveness/Shame *	Assertive	45	8.956	3.914	Student	−2.883	63	0.005	−0.775
		Avoidant	20	12.200	4.764	Mann–Whitney	271.500		**0.011**	−0.397
	Social isolation/Alienation *	Assertive	45	10.133	4.526	Student	−1.701	63	0.094	−0.457
		Avoidant	20	12.300	5.202	Mann–Whitney	334.000		0.099	−0.258
	Failure to achieve*	Assertive	45	9.244	4.370	Student	−2.400	63	0.019	−0.645
		Avoidant	20	12.300	5.497	Mann–Whitney	309.000		**0.044**	−0.313
	Dependency/Incompetence	Assertive	45	10.533	4.071	Student	−2.743	63	**0.008**	−0.737
		Avoidant	20	13.600	4.358	Mann–Whitney	272.500		0.012	−0.394
	Vulnerability to harm	Assertive	45	12.178	5.386	Student	−1.027	63	0.308	−0.276
		Avoidant	20	13.700	5.805	Mann–Whitney	382.000		0.336	−0.151
	Attachment *	Assertive	45	9.022	3.793	Student	−3.918	63	<0.001	−1.053
		Avoidant	20	13.700	5.667	Mann–Whitney	210.500		**<0.001**	−0.532
	Subjugation *	Assertive	45	9.667	4.369	Student	−1.853	63	0.069	−0.498
		Avoidant	20	12.050	5.633	Mann–Whitney	347.000		0.143	−0.229
	Self-sacrifice	Assertive	45	16.956	5.473	Student	−0.100	63	0.920	−0.027
		Avoidant	20	17.100	5.088	Mann–Whitney	441.000		0.904	0.000
	Emotional inhibition	Assertive	45	9.933	3.780	Student	−2.440	63	**0.018**	−0.656
		Avoidant	20	12.800	5.502	Mann–Whitney	314.500		0.054	−0.301
	Unrelenting standards	Assertive	45	15.556	5.446	Student	−0.600	63	0.550	−0.161
		Avoidant	20	16.400	4.706	Mann–Whitney	391.500		0.408	−0.130
	Entitlement/Grandiosity	Assertive	45	13.022	4.901	Student	−2.030	63	**0.047**	−0.546
		Avoidant	20	15.700	4.921	Mann–Whitney	301.000		0.034	−0.331
	Insufficient self-control	Assertive	45	11.267	4.530	Student	−1.784	63	0.079	−0.480
		Avoidant	20	13.600	5.567	Mann–Whitney	333.000		0.097	−0.260

Note: * The test of normality showed significant results (*p* < 0.001). We used the test of Mann–Whitney to analyze differences for these cases. Bold indicates significant differences between groups. All the comparisons were set at *p* ≤ 0.05. For the Student’s *t*-test, the effect size is given by Cohen’s d, and for the Mann–Whitney test, by the rank biserial correlation.

**Table 3 ijerph-18-06627-t003:** Comparatives for coping strategies and early maladaptive schemes scores across the decisions facing body shaming.

Scales	Subscales	Group	Number	Mean	Standard Deviation	Test	Statistic	Degrees of Freedom	*p*	Effect Size
Coping Strategies	Problem-solving	Assertive	48	34.521	8.359	Student	1.044	64	0.301	0.288
		Avoidant	18	32.167	7.587	Mann–Whitney	470.000		0.589	0.088
	Social support seeking	Assertive	48	19.542	7.529	Student	−2.080	64	**0.041**	−0.575
		Avoidant	18	23.667	6.088	Mann–Whitney	283.000		0.032	−0.345
	Wishful thinking *	Assertive	48	19.521	8.968	Student	−1.620	64	0.110	−0.448
		Avoidant	18	23.500	8.665	Mann–Whitney	313.000		0.087	−0.275
	Religious coping	Assertive	48	25.313	8.177	Student	−1.577	64	0.120	−0.436
		Avoidant	18	28.667	6.183	Mann–Whitney	324.500		0.123	−0.249
	Emotional distancing	Assertive	48	22.729	7.629	Student	−1.487	64	0.142	−0.411
		Avoidant	18	25.722	6.229	Mann–Whitney	294.000		0.047	−0.319
	Professional support seeking *	Assertive	48	10.604	5.511	Student	−2.214	64	0.030	−0.612
		Avoidant	18	14.000	5.657	Mann–Whitney	263.500		**0.015**	−0.390
	Overt emotional expression *	Assertive	48	10.604	3.880	Student	−2.953	64	0.004	−0.816
		Avoidant	18	14.111	5.279	Mann–Whitney	261.500		**0.014**	−0.395
	Avoidance coping *	Assertive	48	14.854	4.649	Student	−1.289	64	0.202	−0.356
		Avoidant	18	16.556	5.113	Mann–Whitney	314.000		0.089	−0.273
	Positive reappraisal *	Assertive	48	19.063	4.844	Studen	1.602	64	0.114	0.443
		Avoidant	18	17.000	4.102	Mann–Whitney	519.000		0.211	0.201
	Expression of coping difficulty	Assertive	48	10.479	3.753	Student	−2.430	64	**0.018**	−0.672
		Avoidant	18	12.889	3.085	Mann–Whitney	236.000		0.005	−0.454
	Negative auto-focused coping *	Assertive	48	7.250	2.236	Student	−2.399	64	0.019	−0.663
		Avoidant	18	8.944	3.280	Mann–Whitney	297.500		0.051	−0.311
	Autonomy *	Assertive	48	5.625	2.614	Student	0.730	64	0.468	0.202
		Avoidant	18	5.111	2.349	Mann–Whitney	473.000		0.556	0.095
Early Maladaptive Schemes	Emotional deprivation *	Assertive	48	12.389	5.293	Student	−2.126	64	0.037	−0.588
		Avoidant	18	10.708	4.807	Mann–Whitney	306.000		0.070	−0.292
	Abandonment/Deprivation *	Assertive	48	12.833	5.544	Student	−1.307	64	0.196	−0.361
		Avoidant	18	9.771	4.581	Mann–Whitney	340.500		0.189	−0.212
	Mistrust/Abuse	Assertive	48	11.722	6.153	Student	−1.367	64	0.176	−0.378
		Avoidant	18	12.000	5.336	Mann–Whitney	328.500		0.136	−0.240
	Defectiveness/Shame *	Assertive	48	11.667	4.215	Student	−2.324	64	0.023	−0.642
		Avoidant	18	12.271	5.592	Mann–Whitney	288.500		**0.038**	−0.332
	Social isolation/Alienation *	Assertive	48	13.778	5.082	Studen	−1.534	64	0.130	−0.424
		Avoidant	18	10.188	4.532	Mann–Whitney	331.500		0.148	−0.233
	Failure to achieve *	Assertive	48	13.222	6.422	Student	−1.399	64	0.167	−0.387
		Avoidant	18	9.958	4.767	Mann–Whitney	358.000		0.285	−0.171
	Dependency/Incompetence *	Assertive	48	12.556	5.238	Student	0.238	64	0.812	0.066
		Avoidant	18	17.208	5.375	Mann–Whitney	428.000		0.960	−0.009
	Vulnerability to harm *	Assertive	48	16.222	5.579	Student	−0.998	64	0.322	−0.276
		Avoidant	18	10.563	4.649	Mann–Whitney	353.000		0.257	−0.183
	Attachment *	Assertive	48	11.889	5.016	Student	−2.152	64	0.035	−0.595
		Avoidant	18	15.854	5.535	Mann–Whitney	317.500		0.099	−0.265
	Subjugation *	Assertive	48	14.389	4.565	Student	−1.919	64	0.059	−0.530
		Avoidant	18	14.104	5.337	Mann–Whitney	304.000		0.065	−0.296
	Self-sacrifice	Assertive	48	13.611	4.434	Student	0.657	64	0.513	0.182
		Avoidant	18	11.854	5.036	Mann–Whitney	470.500		0.584	0.089
	Emotional inhibition *	Assertive	48	13.111	4.993	Student	−1.010	64	0.316	−0.279
		Avoidant	18	12.389	5.293	Mann–Whitney	359.000		0.295	−0.169
	Unrelenting standards	Assertive	48	10.708	4.807	Student	1.001	64	0.320	0.277
		Avoidant	18	12.833	5.544	Mann–Whitney	491.000		0.398	0.137
	Entitlement/Grandiosity	Assertive	48	9.771	4.581	Student	0.349	64	0.728	0.096
		Avoidant	18	11.722	6.153	Mann–Whitney	433.500		0.988	0.003
	Insufficient self-control *	Assertive	48	12.000	5.336	Student	−0.905	64	0.369	−0.250
		Avoidant	18	11.667	4.215	Mann–Whitney	371.500		0.386	−0.140

Note: * The test of normality showed significant results (*p* < 0.001). We used the test of Mann–Whitney to analyze differences for these cases. Bold indicates significant differences between groups. All the comparisons were set at *p* ≤ 0.05. For the Student’s *t*-test, the effect size is given by Cohen’s d, and for the Mann–Whitney test, by the rank biserial correlation.

## Data Availability

Data supporting the reported results can be found asking directly of the first author.

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
