# Peer review of "Women Facing Psychological Abuse: How Do They Respond to Maternal Identity Humiliation and Body Shaming?"

_ijerph, 2021, doi:10.3390/ijerph18126627_

Round 1

Reviewer 1 Report

It is an interesting paper and an important topic as well. Very well presented as well. 

However, for the benefit of readers that may not be very familiar with quantitative analysis, I would advise that you first explain what the values (less, and high, negative, and positive,) of the different quantitative analysis result like 'Student’s t and Mann-Whitney test' mean. This will better help all readers to understand and appreciate the article well.

Also, there are few typo errors that you should proofread to rectify.

Author Response

Dear reviewer

We appreciate your comments and suggestions. Below you will find our answers:

Point 1: However, for the benefit of readers that may not be very familiar with quantitative analysis, I would advise that you first explain what the values (less, and high, negative, and positive,) of the different quantitative analysis result like 'Student’s t and Mann-Whitney test' mean. This will better help all readers to understand and appreciate the article well.

Response: We took note of your suggestion and made changes between lines 299-327; 386-388; 415-435 and 476 to 491 to clarify the results.

Point 2: Also, there are few typo errors that you should proofread to rectify.

Response: We made a review of spelling and grammar

Thank you 

Reviewer 2 Report

Abstract

  1. Kindly state precisely the coping strategies of abused women
  2. How do women get social and professional support
  3. Victim cognitive profile: Dont assume that all readers will understand. Be explicit

Introduction

  1. Delete line 164-164 (After all, there is no research with first hand data)

Methodology

  1. Provide the breakdown of how the 73 women were selected from the 7 cities
  2. what type of purposive sampling was used
  3. Kindly state the number of foundation were data were collected from

Result

  1. Kindly recast line 502-509 (Its clumsy)

Conclusion

1. Line 515-516 state briefly the series of coping strategies utilised by women in the study

Author Response

Dear reviewer

We appreciate your comments and suggestions. Below you will find our answers:

Abstract

Point 1. Kindly state precisely the coping strategies of abused women

Response: We included it in abstract and in lines 684-694.

Point 2. How do women get social and professional support –

Response: Social and professional support corresponds to evaluated coping strategies. In this particular case, these women have the support of foundations. Thus, for developing this research, we contacted those foundations to connect with the women. It shows that these women have already resorted to this strategy in the past.

Point 3. Victim cognitive profile: Dont assume that all readers will understand. Be explicit

Response: Done

Introduction

Point 4. Delete line 164-164 (After all, there is no research with first hand data)

Response: The idea was clarified from line 189 to 196.

Methodology

Point 5. Provide the breakdown of how the 73 women were selected from the 7 cities.

Response: Done in lines 208 to 213.

Point 6. What type of purposive sampling was used-

Response: Done in lines 205 and 206.

Point 7. Kindly state the number of foundation were data were collected from.

Response: Done in line 207.

Result

Point 8. Kindly recast line 502-509 (Its clumsy)

Response: Done in lines 672 to 681.

Conclusion

Point 9. Line 515-516 state briefly the series of coping strategies utilised by women in the study

Response: Done

Thank you 

Reviewer 3 Report

Review:

Line specific comments:

48-49 – not always perceived in a particular location? Or in general?

55 – several generally means more than 3 or 4.

57 - To say that something is the most common you would need to be able to cite more than one article that says so.

59-62 – this passage about the identity of a mother is not fully clear.

84 – You introduce the gender parity debate but not fully discuss it. If you are going to mention it make sure you do so fully. You present it as if this assertion of women’s equal violence is fact, when it is a heavy contested conclusion.

162 – Saying that something hasn’t been researched is not enough to say that it is important or significant. You need to say more about why this gap in the knowledge is a concern.

268-269 – I do not know what you mean by ”facilitate the generated knowledge’s social appropriation.”

General Comments:

Several grammar and spelling errors that confuse meaning.

You appear to introduce the topic as intimate partner violence, but then revert to using domestic violence. Be clear in your choice and stick to it. There are ideological reasons behind the different terminologies. Not that you need to go into those here but switching between the two makes it appear that you are not aware.

Be careful of some of your literature. There are some places where you make it sound like aggression from women is the cause of their abuse, which sounds like victim-blaming attitudes. Make sure you present the research findings in a clear way to avoid this.

Throughout the manuscript you use the phrase women victims of domestic violence, which does not read a grammatically correct. You would say either female victims of domestic violence or victims of domestic violence who are women.

You need to address head on the challenge with vignette research. We know that what people think they will do in a violent situation, especially one in the context of a relationship, and what they actually do are not the same. Reported behaviors and actual behaviors are frequently different.

Your hypotheses make it sound like the response to the abuse is what is forming the maladaptive schemes and coping, but later it seems like it is the other way around. You need to be more clear.

A lot of your literature is more focused on the mother and children aspect, which does not explain the focus on body shaming. The way the argument is constructed does not explain why these two types of psychological abuse are included over others.

Author Response

Dear reviewer. 

We appreciate your comments and suggestions, which allow us to improve the paper. Below you will find our answers:

Point 1: 48-49 – not always perceived in a particular location? Or in general? –

Response: The idea was clarified in lines 57-59.

Point 2: 55 – several generally means more than 3 or 4.

Response: The idea was adjusted in line 65

Point 3: 57 - To say that something is the most common you would need to be able to cite more than one article that says so.

Response: The idea was adjusted in line 68

Point 4: 59-62 – this passage about the identity of a mother is not fully clear.

Response: The text was clarified in lines 68 to 72.

Point 5: 84 – You introduce the gender parity debate but not fully discuss it. If you are going to mention it make sure you do so fully. You present it as if this assertion of women’s equal violence is fact, when it is a heavy contested conclusion.

Response: The idea was clarified in lines 94 to 99.

Point 6: 162 – Saying that something hasn’t been researched is not enough to say that it is important or significant. You need to say more about why this gap in the knowledge is a concern.

Response: The idea was clarified and some data about the problem in the participant cities was added (lines 189 to 196).

Point 7: 268-269 – I do not know what you mean by ”facilitate the generated knowledge’s social appropriation.”

Response: The idea was clarified in lines 351 to 356.

General Comments:

Point 8: Several grammar and spelling errors that confuse meaning.

Response: The spelling and grammar were improved.

Point 9: You appear to introduce the topic as intimate partner violence, but then revert to using domestic violence. Be clear in your choice and stick to it. There are ideological reasons behind the different terminologies. Not that you need to go into those here but switching between the two makes it appear that you are not aware.

Response: Much of the literature that we consulted, refers to the study of maladaptive schemes in the context of intimate relationships, and these academic articles do not always appear under the criterion of domestic violence. However, due to the population characteristics and the heteropatriarchal relationships in the participant's experiences, the concept that fits the most is domestic violence. We are not able to determine if the violence has been only from their partners. In some cases, we used "intimate partner violence" into specific situations of aggression that occurs within a relationship.

Point 10. Be careful of some of your literature. There are some places where you make it sound like aggression from women is the cause of their abuse, which sounds like victim-blaming attitudes. Make sure you present the research findings in a clear way to avoid this.

Response: We took note of this, and we improved it.

Point 11. Throughout the manuscript you use the phrase women victims of domestic violence, which does not read a grammatically correct. You would say either female victims of domestic violence or victims of domestic violence who are women.

Response: Done

Point 12. You need to address head on the challenge with vignette research. We know that what people think they will do in a violent situation, especially one in the context of a relationship, and what they actually do are not the same. Reported behaviors and actual behaviors are frequently different.

Response: Indeed, what a person says they would do in a situation does not necessarily correspond to what they actually do if they were exposed to it. For this reason, we decided to analyze participants’ responses to a proposed vignettes and their possible relationship with the cognitive profile of female victims. Maladaptive schemes and coping strategies configure beliefs that can influence attitudes towards certain situations. This study does not seek to predict women’s behavior in situations of violence. However, it does allow us to know their thinking styles, which can influence both their behavior and the consolidation of socially shared beliefs to justify violence and place women in a position of submission.

Point 13. Your hypotheses make it sound like the response to the abuse is what is forming the maladaptive schemes and coping, but later it seems like it is the other way around. You need to be more clear.

Response: We carried out the comparative analysis based on the participants’ decisions to face the proposed situations. This way, through participants’ reaction, women formed the contrast groups for the analysis of their decisions facing the proposed situations. Subsequently, we analyzed their cognitive profiles using coping strategies and maladaptive schemes scales. Our conclusions indicate a dependency relationship between thinking about responding to a situation of violence in a certain way, and the cognitive profile. This is the way we prove our hypotheses. However, we improve the results explanations to clearly portrait the arguments.

Point 14. A lot of your literature is more focused on the mother and children aspect, which does not explain the focus on body shaming. The way the argument is constructed does not explain why these two types of psychological abuse are included over others.

Response: We choose these two evaluations because they focused on different aspects of women self-concept: one is associated to motherhood and the other is related to corporal image acceptance. Both have impact on mental health and have been further explored in psychological studies. We especially present the literature review are about the mother and child relation considering that our main results are based on this situation.

Thank you

Round 2

Reviewer 3 Report

Minor edits:
Line 69 - Change to "one of the main ways"
Line 77 - experiment to experience 
Line 84-86 - clear up the phrasing here "in the people around them" is not clearly tied grammatically to the ideas discussed prior. 
Line 159: change women to women’s

Larger areas:
Line 89-91 - when women respond with violence as a defense mechanism then it is not patriarchal violence as you argue here. Just like someone espousing negative white stereotypes is not reverse racism – as racism is based on a system of power. It’s important that you distinguish between types of IPV – situational couple violence, intimate terrorism, and violent resistance. See work by MP Johnson. You still appear to be framing women’s violent resistance or situational couple violence as equal with intimate terrorism. 
Are children also present during the body shaming? The way you phrase the situations at points throughout the paper it sounds like they are not, but are at other times. This needs to be clarified. And if the children are not present during the body shaming, then the title of the piece needs to change to reflect this. While you state that the important findings are about the presence of children, it seems like the inclusion of body shaming is ancillary or superfluous. 
Sections are required in the literature or study set-up that directly time body shaming with the woman self-concept.

Author Response

Response to Reviewer Comments

Point 1: Line 69 - Change to "one of the main ways"

Response: Done

Point 2: Line 77 - experiment to experience

Response: Done

Point 3: Line 84-86 - clear up the phrasing here "in the people around them" is not clearly tied grammatically to the ideas discussed prior.

Response: Done

Point 4: Line 159: change women to women’s

Response: We clarify this part using another grammatical formula.

Point 5: Larger areas: Line 89-91 - when women respond with violence as a defense mechanism then it is not patriarchal violence as you argue here. Just like someone espousing negative white stereotypes is not reverse racism – as racism is based on a system of power. It’s important that you distinguish between types of IPV – situational couple violence, intimate terrorism, and violent resistance. See work by MP Johnson. You still appear to be framing women’s violent resistance or situational couple violence as equal with intimate terrorism. 

Response: We really appreciate this comment, we were able to read MP Johnson’s work and we used his typology of IPV to better explain the use of violent resistance from female victims of intimate partner violence, particularly intimate terrorism.

Point 6: Are children also present during the body shaming? The way you phrase the situations at points throughout the paper it sounds like they are not, but are at other times. This needs to be clarified. And if the children are not present during the body shaming, then the title of the piece needs to change to reflect this. While you state that the important findings are about the presence of children, it seems like the inclusion of body shaming is ancillary or superfluous. 

Response: No, children are not present during body shaming. We specified in the introduction that we analyzed body shaming by itself, and on the other hand, maternal humiliation while children as witnesses. We added more references and discussing body shaming. Also, we included it in the title.

Point 7: Sections are required in the literature or study set-up that directly time body shaming with the woman self-concept.

Response: Thanks for this recommendation. We add more literature about body shaming and its influence on women’s self-concept.
